# Genome-Wide Identification of Histone Deacetylases and Their Roles Related with Light Response in Tartary Buckwheat (*Fagopyrum tataricum*)

**DOI:** 10.3390/ijms24098090

**Published:** 2023-04-30

**Authors:** Huiling Yan, Hongxu Chen, Qingxia Liao, Mengying Xia, Tian Yao, Lianxin Peng, Liang Zou, Gang Zhao, Jianglin Zhao, Ding-Tao Wu

**Affiliations:** 1Key Laboratory of Coarse Cereal Processing, Ministry of Agriculture and Rural Affairs, Sichuan Province Engineering Technology Research Center of Coarse Cereal Industralization, School of Food and Biological Engineering, Chengdu University, Chengdu 610106, China; yhl201428010715026@163.com (H.Y.); ming981001@163.com (H.C.); cdulqx0326@163.com (Q.L.); xmy2834186005@163.com (M.X.); 13982152430@163.com (T.Y.); penglianxin@cdu.edu.cn (L.P.); zouliang@cdu.edu.cn (L.Z.); zhaogang@cdu.edu.cn (G.Z.); dt_wu@sicau.edu.cn (D.-T.W.); 2Key Laboratory of South China Agricultural Plant Molecular Analysis and Genetic Improvement, South China Botanical Garden, Chinese Academy of Sciences, Guangzhou 510650, China

**Keywords:** Tartary buckwheat, histone deacetylases, light response

## Abstract

Histone deacetylases (HDACs), known as histone acetylation erasers, function crucially in plant growth and development. Although there are abundant reports focusing on *HDACs* of Arabidopsis and illustrating their important roles, the knowledge of *HDAC* genes in Tartary buckwheat (Polygonales Polygonaceae *Fagopyrum tataricum* (L.) Gaertn) is still scarce. In the study, a total of 14 *HDAC* genes were identified and divided into three main groups: Reduced Potassium Dependency-3/His-52 tone Deacetylase 1 (RPD3/HDA1), Silent Information Regulator 2 (SIR2), and the plant-53 specific HD2. Domain and motif composition analysis showed there were conserved domains and motifs in members from the same subfamilies. The 14 *FtHDACs* were distributed asymmetrically on 7 chromosomes, with three segmental events and one tandem duplication event identified. The prediction of the *cis*-element in promoters suggested that *FtHDACs* probably acted in numerous biological processes including plant growth, development, and response to environmental signals. Furthermore, expression analysis based on RNA-seq data displayed that all *FtHDAC* genes were universally and distinctly expressed in diverse tissues and fruit development stages. In addition, we found divergent alterations in *FtHDACs* transcript abundance in response to different light conditions according to RNA-seq and RT-qPCR data, indicating that five *FtHDACs* might be involved in light response. Our findings could provide fundamental information for the HDAC gene family and supply several targets for future function analysis of *FtHDACs* related with light response of Tartary buckwheat.

## 1. Introduction

In Eukaryotes, gene expression can be modulated by epigenetic regulation, which involves the switches between condensed and relaxed states of chromatin structure [1]. Chromatin is formed by DNA and nuclear histone, modifications of which play essential roles in making chromatin into either a transcriptionally active or an inert state [1]. Histone acetylation, as a major form of epigenetic modification, is a reversible biological process mediated by histone acetyltransferases (HATs) and histone deacetylases (HDACs) functioning on the ε-amino group of conserved lysine residues (such as H3K9, H4K5, H2AK5, H2BK5 and so on) within the core histones [2]. In general, the positive charge of histone tails can be neutralized by the acetylation of lysine residues [1,3,4]. Therefore, histone acetylation catalyzed by HATs can reduce the affinity of histone tails for negatively charged DNA strands, resulting in the loosening of chromatin, which benefits the formation of binding sites for transcription factors and active gene expression [5,6,7]. In turn, histone deacetylation catalyzed by HDACs might boost the interaction between histone tails and DNA strands, leading to chromatin condensation and gene silencing and repression [4].

HDACs are widespread in eukaryotes and can be classified into three main subfamilies in plants basing on sequence homology with yeast HDACs and phylogenetic analysis [3,8]. The HDACs in plant are composed of the RPD3/HDA1, SIR2, and HD2 subfamilies [9], among which the RPD3/HDA1 and SIR2 subfamilies require a zinc ion and nicotine adenine dinucleotide (NAD) as a cofactor, respectively [10,11], while the HD2 subfamily is considered to be zinc-dependent [12]. In plants, HDACs play crucial roles in genome stability, gene expression, plant growth, development, biotic and abiotic stress responses such as seed development and dormancy, seed germination, seedling growth, leaf development, root development, flowering-time control, fruit development and ripening, and fruit senescence, as well as responses to hormones and environmental stresses [9,13,14,15,16,17,18,19,20,21,22,23,24].

Plant growth and development is an integrated result of intrinsic and extrinsic signals, of which light is one of the most significant environmental factors [25]. In order to trigger various photomorphogenic processes under different developmental conditions, plants have evolved a sophisticated photosensory system, in which the wavelength, intensity, direction, and duration of light is perceived by a set of photoreceptors [26,27]. In these processes, the induced transcription activation of light responsive nuclear genes by the photosensory system has been proven as a significant change in light response. In addition, histone acetylation, a state regulated by HATs and HDACs, has been proven to be involved in light-induced gene activation [28,29]. For example, the mutation of *HAF2*, a HAT, decreased H3 acetylation on light-responsive promoters and light-induced mRNA levels, resulting in reduced chlorophyll accumulation [30]. The loss-of-function mutant of *GCN5*, a HAT, resulted in reduced H3 and H4 acetylation on target loci and induced early light-responsive genes under light conditions, and these worked together to generate a long hypocotyl phenotype [31,32]. In the dark, HDA15 interacts with PIF3 to co-repress the light-responsive genes via decreasing H4 acetylation and RNA Polymerase II-related transcription, leading to reduced chlorophyll biosynthesis and photosynthesis [33]. In the light, HDA15 interacts with NF-YCs (NF-YC1, NF-YC3, NF-YC4, and NF-YC9) to co-repress the expression of a set of hypocotyl elongation-associated genes via H4 deacetylation, producing decreased hypocotyl elongation along the early seedling stage [34]. However, this is just a small part of the sophisticated regulatory network, and the role of histone acetylation is still largely uncovered, especially in non-model species.

Tartary buckwheat (*F. tataricum*), a well-known pseudocereal crop belonging to the genus Fagopyrum of the family Polygonacea and tolerant to adverse environments, is generally cultivated in mountainous alpine areas, where there are harsh climates and low-nutrient soils [35]. *F. tataricum* is popular as a significant medicine and health food due to its rich content of bioactive flavonoids, which are detected to be highly beneficial for human health [36]. In recent years, studies have characterized a set of modulators playing crucial roles in the growth and development of *F. tataricum* under certain environmental factors [37,38,39,40,41,42,43]. For example, FtMYB6 and FtMYB116, were found to be positive regulators of flavonoids biosynthesis in the light [37,38]. The FtMYB9, FtMYB13, FtbZIP5, and FtbZIP83 were identified as positive regulators of drought and salt response and improved drought and salt tolerance of transgenic Cruciformes Cruciferae *Arabidopsis thaliana* (L.) Heynh., while FtMYB10 was found to be a negative regulator of drought and salt response in transgenic *Arabidopsis thaliana* [39,40,41,42,43]. However, previous studies mainly focused on the transcriptional and post-transcriptional levels [37,38,39,40,41,42,43], and thus the knowledge of the pre-transcriptional level (epigenetic level) is extremely limited.

In this study, the genome-wide analysis of *HDACs* was conducted on the *F. tataricum* genome. We investigated chromosomal localization, gene structure, conserved motifs, phylogenetic relationships, and putative *cis*-elements in the promoters of *FtHDAC* genes. In addition, the expression patterns of *FtHDACs* in various tissues, fruits of different development stages, and seedlings under different light conditions were analyzed. The objective of the current study was to shed light on potential roles of *FtHDACs* in *F. tataricum* growth, development, and light response.

## 2. Results

### 2.1. Identification and Characterization of HDACs in F. tataricum

The HDACs protein sequences in Cruciformes Cruciferae *Arabidopsis thaliana* (L.) Heynh. and Cruciformes Cruciferae *Oryza sativa* L. were employed as queries to search against the Tartary buckwheat genome through the BLASTP program. In total, 14 FtHDAC genes were identified from the *F. tataricum* genome after eliminating the redundant sequences. A phylogenetic analysis of HDAC families in Arabidopsis, rice, and Tartary buckwheat was performed to analyse the evolutionary relationships and classification of FtHDACs (Figure 1). The results demonstrated that 14 FtHDACs were divided intoto three subfamilies as *Arabidopsis thaliana* and *Oryza sativa*, with 9 members in the RPD3/HDA1 subfamily, 2 members in the SIR2 subfamily, and 3 members in the HD2 subfamily. Furthermore, we renamed the *FtHDAC* genes according to their homologs in *Arabidopsis thaliana*. The protein properties analysis displayed that the protein lengths of the 14 FtHDACs were distributed a broad range, spanning from 183 AA (FtHDA5) to 502 AA (FtHDA19) (Table 1). As a result, FtHDA5 exhibited the smallest predicted molecular weight (MW) (20.68 kDa), while FtHDA19 presented the biggest predicted MW (56.19 kDa). In addition, the predicted isoelectric point (pI) distributed from 4.11 (FtHDT1) to 9.39 (FtHDA5).

### 2.2. Motif and Domain Composition of the FtHDAC Genes

The putative motifs of the FtHDACs were detected via the program MEME (Figure 2A). Thirty conserved motifs differing in length and amino acid component were identified and named: from motif 1 to motif 30. There were at least two motifs present in each subfamily, suggesting that they belonged to the same family. Motifs 1 and 3 were specific to the RPD3/HDA1 subfamily members. Motifs 11, 12, 14, 17, 19, 20 and 21 were specific to SIR2 proteins, while motifs 9, 13 and 24 were specific to HD2 proteins. The conserved domains were further detected with SMART and HMMER, and visualized via IBS software (version 1.0.3) (Figure 2B). There was a deacetylase catalytic domain (HDAC domain) in all RPD3/HDA1 subfamily members. The members of the SIR2 subfamily all contained a SIR2 domain. A Nucleoplasmin-like domain (NPL) was present in the three members of the HD2 subfamily. Furthermore, there was a conserved ZnF_C2H2 domain in FtHDT2 and FtHDT3 but not in FtHDT1. 

### 2.3. Phylogenetic Analysis and Gene Structure of the FtHDAC Gene Family

The protein sequences of FtHDACs were aligned and applied for the Neighbor-Joining (NJ) phylogenetic tree construction via MEGA6.06. As shown in Figure 3A, there were three subgroups in the tree, with member names in three different colors (RPD3/HDA1, black; SIR2, red; HD2, blue). There were 3, 1, and 1 major groups in the RPD3/HDA1, SIR2, and HD2 branch, respectively. As for the first group of the RPD3/HDA1 branch, the paralogous RPD3/HDA1 pair consisted of FtHDA8-1 and FtHDA8-2 on the outermost side and FtHDA14 and FtHDA2 located inside. Diversely, the paralogous RPD3/HDA1 pair consisted of FtHDA6-1 and FtHDA6-2, located on the innermost side, and FtHDA9 and FtHDA19 located outside in the second group. Moreover, there was only one member (FtHDA5) in the third group. Additionally, there was one paralogous pair formed by FtSRT1 and FtSRT2 in the SIR2 branch, and there was a paralogous HD2 pair consisting of FtHDT2 and FtHDT3 with FtHDT1 distributed outside in the HD2 branch. The gene structure was displayed by the TBtools software (version 1.108) to demonstrate the structures of the *FtHDACs* (Figure 3B). As shown in Figure 3B, the gene structures of *FtHDACs* were diverse, and the number of introns in *FtHDACs* varied from two to thirteen. As for members of the SIR2 subfamily, the gene structures were conserved and there were thirteen introns present in each SIR2 gene. In addition, the data displayed that *FtHDT1*, *2*, and *3* contained seven, nine, and seven introns, respectively. Taken together, the intron/exon distribution pattern was varying in members of the RPD3/HDA1 and HD2 subfamilies, while that of the SIR2 subfamily was conserved.

### 2.4. Chromosomal Location and Duplications of the FtHDACs in the F. tataricum Genome

Chromosomal location of *FtHDACs* in the *F. tataricum* genome was carried out via the TBtools program (Figure 4). The results illustrated that *FtHDACs* were located asymmetrically on seven *F. tataricum* chromosomes and no *FtHDAC* was distributed on chromosome Ft2. Chromosome Ft6 contained the highest density of *HDACs*, with four members (*FtHDA8-1*, *FtHDA8-2*, *FtHDA14* and *FtHDA19*), followed by chromosome Ft1 with three genes (*FtHDA6-1*, *FtHDT1*, and *FtSRT2*) and chromosome Ft4 with two genes (*FtHDA6-2* and *FtHDA5*), while there was only one *HDAC* gene present on each of the chromosomes Ft3, Ft5, Ft7 and Ft8. Four paralogous *HDAC* gene pairs were detected in *F. tataricum* according to the phylogenetic tree of FtHDACs (Figure 3A). The data illustrated three duplications consisting of genes from distinct chromosomes and one duplication consisting of genes (*FtHDA8-1* and *FtHDA8-2*) distributed at the same chromosome, which was characterized as a tandem duplication since the two homologous genes were located on the chromosome Ft6 within a 20 kb physical distance (7061 bp) (Table 1) with 70.6% and 69.63% nucleotide and protein sequence similarity, respectively. To analyze the evolutionary selection process of the *FtHDAC* gene family, the ratio of Ka/Ks was computed via the TBtools program (Table 2). Notably, there was no result for Ks of the gene pair consisting of *FtHDT2* and *FtHDT3* since the sequence divergence value (0.83) was higher than 0.75. As for the other three gene duplication events, the Ka/Ks ratios were less than one, indicating that negative selection was the main driving force for the expansion of *FtHDACs*. To track the dates of duplication events, their approximate dates were computed. As shown in Table 2, the two segmental duplication events of the *FtHDACs* originated from 38.77 to 88.57 million years ago, while the tandem duplication of *FtHDA8-1* and *FtHDA8-2* occurred 2.47 million years ago. 

### 2.5. Prediction of Cis-Regulatory Elements in the Promoter Regions of the FtHDACs

Up-to-2-kb upstream regions of the *FtHDACs* were analyzed with the Plant Care server to detect probable *cis*-regulatory elements. There were in total 22 *cis*-regulatory elements found in the promoters of *FtHDACs* (Figure 5). More than six types of *cis*-regulatory elements existed in the promoters of 13 *FtHDACs*, while there were only three types of *cis*-regulatory elements present in the promoter of *FtHDA14*. Among the identified *cis*-regulatory elements, the element related to light response was the most abundant and present in the promoters of all the *FtHDACs* but *FtHDA14*, followed by the element related to MeJA-responsiveness which was present in the promoters of 12 *FtHDACs* but not *FtHDA2* and *FtHDA14*. As for the *cis*-regulatory elements involved in abscisic acid, salicylic acid, gibberellin, and auxin response: these were present in promoters of 9, 3, 5, and 5 FtHDACs, respectively. There was a *cis*-regulatory element related to anaerobic induction present in the promoters of 10 *FtHDACs*. In addition, the promoters of 3, 3, 3, 5, 6, 6, 7, and 7 *FtHDACs* contained the *cis*-regulatory element involved in zein metabolism regulation, MYBHv1 binding site, enhancer-like element involved in anoxic-specific inducibility, cis-acting regulatory element related to meristem expression, MYB binding site involved in drought-inducibility, cis-acting regulatory element involved in circadian control, low-temperature responsiveness, and defense and stress responsiveness, respectively. The *cis*-regulatory element involved in endosperm expression, wound-responsive element, and binding site of AT-rich DNA binding protein (ATBP-1) only existed in the promoters of 1or 2 *FtHDACs*. These results indicated that FtHDACs might be involved in not only plant-environment interaction, but also phytohormone and stress signaling. 

### 2.6. Tissue-Specific Expression Profiles of the FtHDACs

To enrich the knowledge of potential roles, the transcript abundance of *FtHDACs* in root, stem, leaf, flower, and fruit of three diverse developmental phases was extracted from the Tartary Buckwheat Database with gene IDs and utilized to build a heat map (Appendix A). The data displayed that 14 *FtHDACs* were expressed at different levels in distinct tissues, which suggested that *FtHDACs* might play various roles in plant growth and fruit development in *F. tataricum*. The transcript of 14 *FtHDACs* could be detected in all the abovementioned tissues. Among these *FtHDACs*, *FtHDT1* showed the highest abundance of transcript, while *FtHDA6-1* was expressed at extremely low levels in all the detected tissues. Five *FtHDACs* (*FtHDA2*, *FtHDA5*, *FtHDA9*, *FtHDA19*, and *FtHDT1*) and seven *FtHDACs* (*FtHDA6-1*, *FtHDA6-2*, *FtHDA8-1*, *FtHDA8-2*, *FtSRT2*, *FtHDT2*, and *FtHDT3*) illustrated the most abundant transcript levels in roots and fruits at different stages, respectively. In addition, *FtHDA14* and *FtSRT1* exhibited the highest transcript abundance in leaf and flower, respectively. On the other hand, five (*FtHDA6-1*, *FtHDA8-2*, *FtHDA9*, *FtHDA14*, and *FtSRT1*) and four (*FtHDA6-2*, *FtHDA8-1*, *FtSRT2*, and *FtHDT2*) *FtHDACs* were expressed at their lowest levels in stem and leaf, respectively. Four *FtHDACs* (*FtHDA5*, *FtHDA19*, *FtHDT1*, and *FtHDT3*) displayed the lowest transcript levels in fruits at different stages, while only *FtHDA2* was expressed at the least level in flowers. Interestingly, the transcript levels of all the *FtHDACs* showed a decreased tendency from Fruit-13 to Fruit-19, followed by subsequent decrement or increment in 8 (*FtHDA5*, *FtHDA6-1*, *FtHDA8-1*, *FtHDA8-2*, *FtHDA14*, *FtSRT1*, *FtSRT2*, and *FtHDT2*) and 6 (FtHDA2, *FtHDA6-2*, *FtHDA9*, *FtHDA19*, *FtHDT1*, and *FtHDT3*) *FtHDACs* from Fruit-19 to Fruit-25, respectively. Altogether, these results demonstrated that the expression profiles of the *FtHDACs* in distinct tissues were overlapping but unique, indicating that *FtHDACs* might play essential roles in sundry physiological processes in *F. tataricum*.

### 2.7. Expression Profile of the FtHDACs in Response to Diverse Light Condition

Light is one of the most significant environmental factors and triggers diverse photomorphogenic events with a set of light-dependent alterations to regulate the whole plant growth and development process. The above *cis*-regulatory element prediction (Figure 5) indicated that *FtHDACs* might be involved in light response, and so we investigated the probable functions of *FtHDACs* in response to diverse light wavelength and duration time with transcriptome or RT-qPCR data. The transcriptome data of *F. tataricum* seedlings treated by diverse light wavelengths (Darkness, Blue, Far-red, and Red) illustrated that the expression levels of the *FtHDACs* varied considerably when responding to diverse light wavelengths (Appendix A) [37]. Compared to the transcript levels of *FtHDACs* under darkness, eight *FtHDACs* (*FtHDA2*, *FtHDA5*, *FtHDA6-1*, *FtHDA8-1*, *FtHDA19*, *FtSRT1*, *FtSRT2*, and *FtHDT1*) showed fluctuated expression patterns under different lightwavelengths, while four (*FtHDA6-2*, *FtHDA9*, *FtHDT2*, and *FtHDT3*) and two (*FtHDA8-2* and *FtHDA14*) *FtHDACs* were inhibited or induced by diverse light wavelengths, respectively. Specifically, blue, far-red, and red light downregulated the expression of nine, six, and eight *FtHDACs*, while five, eight, and six *FtHDACs* were upregulated by blue, far-red and red light, respectively. *FtHDA6-2*, *FtHDA9*, *FtHDT2* and *FtHDT3* were expressed at their highest expression levels under darkness, while *FtHDA14* and *FtSRT1* were most intensely induced by blue light. *FtHDA2*, *FtHDA5*, *FtHDA8-2* and *FtHDA19* displayed the most abundant transcripts under far-red light treatment, while red light induced the transcripts of *FtHDA6-1*, *FtHDA8-1*, *FtSRT2*, and *FtHDT1* to the most abundant extent. 

Furthermore, the relative expression levels of *FtHDACs* under different light duration conditions were analyzed by RT-qPCR with two reference genes to investigate the role of *FtHDACs* in the photomorphogenic process under white light treatment (Figure 6 and Appendix A). Integratedly, the relative expression levels of *FtHDACs* fluctuated considerably along the light treatment. There was a slight difference between the relative expression levels of *FtHDACs* derived from *FtH3* (Figure 6) and *FtACTIN7* (Appendix A), and the expressions similarly and significantly changed under both reference genes were called significant changes. After one hour of white light treatment, only the expression of *FtHDA8-2* was significantly inhibited. With the duration of white light treatment, three *FtHDACs* (*FtHDA6-1*, *FtHDA8-1*, and *FtHDA8-2*) were obviously inhibited at 2 h, while three (*FtHDA6-1*, *FtHDA9*, and *FtHDA19*) and one (*FtHDA6-2*) *FtHDACs* were notably induced or inhibited at 6 h, respectively. After 12 h of white light treatment, four *FtHDACs* (*FtHDA14*, *FtHDA19*, *FtHDT2*, and *FtSRT2*) were markedly upregulated, while *FtHDA6-1* was strongly downregulated. When it came to 24 h, *FtHDA6-1*, *FtHDT1*, and *FtSRT2* were inhibited, while only *FtHDA19* was induced. Similarly, only *FtHDA19* was induced at 48 h, but five *FtHDACs* (*FtHDA8-1*, *FtHDA9*, *FtHDA14*, *FtHDT1*, and *FtSRT2*) were clearly inhibited. For specific *FtHDACs*: *FtHDA19* was obviously induced at 6 h, 12 h and 48 h, and *FtHDT1* was notably induced at 24 h and 48 h, while *FtHDA6-1* was obviously inhibited at 2 h, 12 h, and 24 h, *FtHDA8-1* was inhibited at 2 h and 48 h, and *FtHDA8-2* was inhibited at 1 h and 2 h. In addition, *FtSRT2* and *FtHDA14* were induced at 12 h, but inhibited at 24 h and 48 h, respectively. Jointly, the distinct expression patterns of *FtHDACs* under diverse light condition suggested that *FtHDACs* were critical for the light response of *F. tataricum*.

## 3. Discussion

The histone octamer, consisted of two replicates of H2A, H2B, H3, and H4, is identified to wrap DNA to form nucleosome of chromatin. There are varied post-translational modifications (PTMs), such as acetylation, methylation, and phosphorylation, present in the histone tails out of nucleosome [44]. Histone modifications could mediate chromatin structure and/or regulatory factor recruitment, resulting in regulation during DNA replication, DNA repair, and RNA transcription [44]. *HDACs*, called erasers for histone acetylation, are identified to play critical roles in plant growth and plant-environment interaction via gene expression regulation [44]. There are 18 *HDAC* members (14 RPD3/HDA1, 2 SIR2, and 2 HD2) in Cruciformes Cruciferae *Arabidopsis thaliana* (L.) Heynh., 18 *HDAC* members (14 RPD3/HDA1, 2 SIR2, and 2 HD2) in Cruciformes Cruciferae *Oryza sativa* L., 15 *HDAC* members (10 RPD3/HDA1, 2 SIR2, and 3 HD2) in Solanales Solanaceae *Solanum lycopersicum* L., 13 *HDAC* members (10 RPD3/HDA1, 2 SIR2, and 1 HD2) in Vitales Vitaceae *Vitis vinifera* L., 18 *HDAC* members (8 RPD3/HDA1, 4 SIR2, and 6 HD2) in Parietales Theaceae *Camellia sinensis* (L.) O. Ktze., 20 *HDAC* members (16 RPD3/HDA1, 2 SIR2, and 2 HD2) in Cruciformes Cruciferae *Brassica rapa* L., and 28 *HDAC* members (18 RPD3/HDA1, 4 SIR2, and 6 HD2) in Rosales Fabaceae *Glycine max* (L.) Merr. [9,45,46,47,48,49,50]. In the study, 14 *HDAC* members were identified and characterized in *F. tataricum* (Table 1). This number is similar to that of *Arabidopsis thaliana*, *Oryza sativa*, *Solanum lycopersicum,* and *Vitis vinifera*. The 14 *FtHDACs* were divided to three subfamilies—RPD3/HDA1, HD2, and SIR2—and unevenly distributed on all the *F. tataricum* chromosomes except for chromosome Ft2 (Figure 1 and Figure 4). There were nine FtHDACs present in the RPD3/HDA1 family with distinct motif 1 and motif 3, and a symbolic histone deacetylase catalytic domain (Figure 1 and Figure 2). In addition, there were two and three members in the HD2 and SIR2 subfamilies, respectively. As for members of the HD2 and SIR2 subfamilies, they contained no sequence homology with other RPD3/HDA1 family members (Figure 1 and Figure 2). Individually, there was a SIR2 domain in FtSRT1 and FtSRT2, while NPL domains were present in all members of HD2 subfamily, but the ZnF-C2H2 domain only existed in FtHDT2 and FtHDT3 (Figure 1 and Figure 2). These results indicated that there was similarities and differences among the members of the same subfamilies in terms of conserved domains and motifs of HDACs in *F. tataricum*, which were in accordance with those of HDACs in other plants, for instance *Arabidopsis thaliana*, *Solanum lycopersicum,* and *Glycine max*. 

Plant *HDACs* have been identified to play crucial roles in numerous biological processes, such as seed germination, hypocotyl elongation, organ development, flowering, leaf senescence, and stress response [44]. In our study, the potential *cis*-regulatory elements in promoters of *FtHDACs* illustrated that there were diverse *cis*-regulatory elements involved in light response, hormone response, and stress response existing in promoters of most *FtHDACs*, suggesting *HDACs* might act in plant growth, development, and plant-environment interaction. In particular, the *cis*-element involved in light response was the only *cis*-element existing in the promoters of all the *FtHDACs*, indicating the essential role of *FtHDACs* in the light response of *F. tataricum* (Figure 5). Furthermore, transcriptome data downloaded from the Tartary buckwheat database was used to investigate the tissue-specific expression patterns of all the *FtHDACs* (Appendix A). *FtHDA6-1* exhibited extremely low expression levels in all the detected tissues (Appendix A), which seemed contradictory to its homologous gene *AtHDA6*. *AtHDA6* has been identified to participate in numerous biological processes of plant growth and development [44]. Notably, there was a paralogous gene of *FtHDA6-1* in *F. tataricum*, which was expressed at abundant levels in all the detected tissues (Figure 1 and Appendix A). Basing on the close evolutionary relationship and similar expression pattern of *FtHDA6-2* and *AtHDA6*, *FtHDA6-2* might play similar roles in *F. tataricum’s* growth and development as *AtHDA6*. Jointly, these results indicated that there might be function redundancy of paralogous genes (*FtHDA6-1* and *FtHDA6-2*) in *F. tataricum*. The *FtHDA5*, the closest homolog of *AtHDA5*, was expressed at relatively abundant levels in root and flower (Figure 1 and Appendix A), indicating it might be involved in root and flower development. The *FtHDA19* was homologous to *AtHDA19* and *OsHDA702* (Figure 1), which were characterized to play a part in embryonic and flower development, and root development, respectively [44,51]. The *FtHDA19* was expressed at relatively abundant levels in root, stem, and flower (Appendix A), suggesting its potential role in root and flower development. The *FtHDT1*, *FtHDT2* and *FtHDT3* were expressed at their highest levels at Fruit_13 among the three fruit development stages (Appendix A), similar to their homologous genes: *AtHDT1*, *AtHDT2,* and *AtHDT4* (Figure 1) [5,52]. *AtHDT1*, *AtHDT2,* and *AtHDT4* were reported to be involved in young siliques development [52]; thus, *FtHDT1*, *FtHDT2,* and *FtHDT3* might also act in the fruit development of *F. tataricum*. In addition, combining *FtHDA2* was significantly induced from Fruit_19 to Fruit_25 and its homolog: *AtHDA2* was expressed in developing embryos and dry seeds (Figure 1) [53], and so we could hypothesize that *FtHDA2* might be involved in embryo development. AtHDA19 has been identified to interact with SCARECROW-LIKE15 or HSL1 to repress seed maturation [54,55]. The *FtHDA19*, the homolog of *AtHDA19*, was downregulated along with fruit development (Figure 1 and Appendix A), suggesting it might also play a negative role in the seed maturation of Tartary buckwheat. Moreover, unlike *AtHDA8*, which was induced in late-stage seeds [53], *FtHDA8-1* and *FtHDA8-2* were expressed with a decreased tendency in the fruit development of *F. tataricum* (Appendix A). Taken together, the functions of *FtHDACs* might be partially conservative and diverse compared to those of other plants. 

*F. tataricum* is planted mostly in mountainous alpine areas with altitudes of more than 2500 m, where the light condition is distinct from that in the plains. The light condition is crucial for the phenotype and flavonoid biosynthesis of *F. tataricum* [37]. In the current study, we found that the expression of all the *FtHDACs* was altered by different light wavelengths (Appendix A). Furthermore, their expression was diverse under different light treatment time (Figure 6). These data indicated *FtHDACs* might play an essential role in the light response of *F. tataricum*. A deacetylase complex formed by AtHDA19 and SNL1–SNL6 was recruited by HY5 to the chromatin regions of HY5 and BBX22, and thus reduced the accessibility and histone acetylation and repressed their expression, resulting in inhibited photomorphogenesis. Specifically, the *hda19-2* mutant exhibited enhanced photomorphogenesis under far-red, red, and blue light treatments compared with the wild type [56]. *FtHDA19* was downregulated by red and blue light, while far-red light slightly induced its expression (Appendix A). Moreover, *AtHDA19* were inhibited from dark to white light in wild type Arabidopsis seedlings [56], but *FtHDA19* was significantly induced by 6 h, 12 h, and 48 h of white light treatment (Figure 6). These results suggested that the function of *FtHDA19* in light response is partly diverse from that of *AtHDA19*. AtHDA15 has been reported to be recruited by HY5 or NF-YCs to decrease the histone acetylation levels of hypocotyl cell elongation related genes, resulting in inhibited hypocotyl elongation. Phenotypic analysis displayed that AtHDA15 played a negative role in hypocotyl cell elongation under red and far-red light treatments in Arabidopsis seedlings [34,57]. *FtHDA5*, the closest homolog of *AtHDA15*, was markedly induced by far-red and blue light and slightly inhibited by red light (Appendix A), suggesting its potential role in light response. The AtHDA6 was characterized as a positive regulator of light-regulated chromatin compaction. The *athda6* mutant displayed lower chromatin compaction and reduced methylation levels of DNA and histone H3K9Me2 at Nucleolar Organizing Regions [58]. *FtHDA6-2* was repressed by far-red, red, and blue light, while *FtHDA6-1* was repressed by far-red and blue light but induced by red light (Appendix A). However, the transcript levels of *FtHDA6-1* were also extremely low under different light conditions (Appendix A), and we could speculate that there was function redundancy between FtHDA6-1 and FtHDA6-2 in light response of Tartary buckwheat. Jointly, FtHDA6-2 might act in a similar way to AtHDA6 to positively regulate the chromatin compaction of the light response process. These data were in accordance with the findings that HDACs play critical roles in plant growth and development related with light response [34,56,57].

## 4. Materials and Methods

### 4.1. Plant Materials and Treatments

*F. tataricum* seeds (Cultivar Xiqiao No. 1 of *F. tataricum*) were retrieved from the Coarse Cereal Processing Center of Chengdu University (Chengdu, China). The seeds were immersed for 18–22 h in 0.2% sodium chloride solution in darkness after being rinsed in deionized water (dH_2_O) until the liquid was transparent. Afterwards, the seeds were sown on filter paper moistened with liquid medium containing 0.083 g/L of nitrogen fertilizer, 0.083 g/L of potassium fertilizer, and 0.166 g/L of phosphate fertilizer, and were kept in darkness for 4 days at 22 °C. Thereafter, the seedlings were illuminated with light-emitting diodes (LEDs) with the following intensities: 0 and 50 μmol m^−2^ s^−1^, respectively. The samples after 0, 1, 2, 6, 12, 24, and 48 h of light treatments were frozen with liquid nitrogen and kept at −80 °C for future analysis.

### 4.2. Identification and Classification of FtHDAC Genes

The protein sequences of the *Arabidopsis thaliana* and *Oryza sativa* HDACs were obtained from the Arabidopsis Information Resource [59] and the Rice Genome Annotation Project [60], respectively. FtHDACs were identified in Tartary buckwheat genome, obtained from the Tartary Buckwheat Genome Project [35], via the BLASTP method using the *Arabidopsis thaliana* and *Oryza sativa* HDACs as query sequences. The conserved domains of non-redundant FtHDACs were detected by the online NCBI conserved domain searches [61], and the online SMART (version 9) [62] and HMMER (version 3.3.2) [63] softwares. The online ExPASy tool [64] were applied to predict the protein sequence length, molecular weight (MW), and isoelectric point (pI) of the FtHDACs.

### 4.3. FtHDACs Protein Motif, Domain and Gene Structure Analysis

The MEME (Multiple Em for Motif Elicitation) program [65] was used to discover the conservative motifs of the FtHDACs. The domain architecture of FtHDACs was predicted with SMART (version 9) [62] (http://smart.embl-heidelberg.de/accessed on 10 December 2021) and HMMER (version 3.3.2) [63] and visualized via the IBS (Illustrators of Biological Sequences) software (version 1.0.3) [66]. The TBtools software (version 1.108) was applied to visualize the gene structure of the *FtHDACs*.

### 4.4. Phylogenetic Analysis, Genome Distribution and Gene Duplication Analysis of FtHDACs

All the HDAC protein sequences from *F. tataricum*, *Arabidopsis thaliana*, and *Oryza sativa* were aligned using ClustalW and used for phylogenetic analysis with the MEGA 6.06 software in the Neighbor-Joining model with 1000 replicated bootstrap values. The initial chromosomal positions of the *FtHDACs* and chromosome lengths of *F. tataricum* were retrieved from the *F. tataricum* genome, obtained from the Tartary Buckwheat Genome Project [35], and the TBtools software(version 1.108) was employed to visualize the chromosome localization of the *FtHDACs* [30]. The parameters (Ks and Ka: synonymous and nonsynonymous substitution rate, respectively) of the duplication events were analyzed by the TBtools software (version 1.108). The Ka/Ks ratio of homologous gene pairs was applied to define the type of selection they were under. When Ka/Ks is greater than one, genes are under positive selection. A Ka/Ks less than one indicates negative selection. When Ka/Ks is equal to one, genes are under neutral selection. The time (T) of the duplication events was computed with the formula T = Ks/(2 × 1.5 × 10^−8^) Mya [31]).

### 4.5. Prediction and Analysis of Cis-Regulatory Elements in the Promoter Regions of FtHDAC Genes

The 2.0 kb upstream sequences of the transcription start site of *FtHDACs* were retrieved from the Tartary Buckwheat Genome Project [35] and subjected to the online PlantCARE database [67] to detect *cis*-regulatory elements in the putative promoter regions. The identified *cis*-regulatory elements were visualized with the TBtools software (version 1.108).

### 4.6. Transcriptome Data Analysis

The transcript profiles of *FtHDAC* genes in root, stem, leaf, flower, and fruits from diverse development stages of *F. tataricum* were retrieved from the Tartary Buckwheat Database [68]. The transcript profiles of *FtHDAC* genes in seedlings of *F. tataricum* under different light wavelengths (red light (670 nm), blue light (470 nm), and far-red light (735 nm)) were downloaded from Zhang et al. [37]. The RPKM (Reads Per Kilobase of exon model per Million mapped reads) values were transformed to log_2_ (value + 1) and visualized by the TBtools software (version 1.108).

### 4.7. RNA Isolation and Quantitative Real-Time PCR Analyses (RT-qPCR)

The total RNA of *F. tataricum* seedlings after light treatments was extracted with the RNAprep Pure Plant Plus Kit (TIANGEN BIOTECH, Beijing, China) according to the manufacturer’s command. TransScript^®^ All-in-One First-Strand cDNA Synthesis SuperMix for qPCR (One-Step gDNA Removal) (Trans, Beijing, China) was employed to synthesise the first strand of cDNA with the extracted RNA as a template. RT-qPCR primers were devised via the Primer Premier 5 software with the corresponding sequences of *FtHDACs* downloaded from the *F. tataricum* genome. The expression patterns of all the *FtHDACs* after light treatment were investigated by RT-qPCR with *FtH3* and *FtACTIN7* as the internal reference genes based on their similar expression levels in nearly all the tissues. RT-qPCR was performed with TransStart^®^ Top Green qPCR SuperMix (Trans, Beijing, China) via the qTOWER3G system (Analytik Jena GmbH, Jena, Germany) as described by Yan et al. [69]. The relative expression levels of *FtHDAC* genes were calculated with the formula 2^−ΔΔCT^. A total of three independent biological replicates were utilized in the investigation. The primers applied for RT-qPCR are listed in Appendix A.

### 4.8. Statistical Analysis

Experiments were performed in a totally randomized plan with three biological replicates. Data were demonstrated as the mean ±standard error (SE). Data were tested by Student’s *t* test using SPSS version 26.0 (SPSS Inc., Chicago, IL, USA).

## 5. Conclusions

In the current study, in total 14 histone deacetylase genes, consisting of 9, 2, and 3 members of the RPD3/HDA1, SIR2, and HD2 subfamilies, respectively, were identified and characterized. There were conservative domains and motifs in members of the same subfamilies. *FtHDACs* were distributed unevenly on all the chromosomes except chromosome Ft2, with overlapping and distinct genres and numbers of *cis*-regulatory elements in their promoter regions. Negative selection was the major driving force for HDAC gene expansion in *F. tataricum*. The 14 *FtHDACs* were expressed in all the detected tissues with diverse expression profiles, indicating that FtHDACs might be involved in the regulation of various aspects of *F. tataricum* growth and development. Furthermore, study of expression pattern under different light conditions demonstrated that two induced *FtHDACs* (*FtHDA19* and *FtHDT1*) and three inhibited *FtHDACs* (*FtHDA6-1*, *FtHDA8-1* and *FtHDA8-2*) might be crucial regulators of light response in *F. tataricum*. In summary, a potential working model for FtHDACs in the light response of *F. tataricum* is proposed (Figure 7). Our results will be helpful for expanding our understanding of the participation of epigenetic regulators—histone modifiers—in the regulation of light response and supplying some targets for future function and molecular mechanism investigation of HDACs in response to different light conditions.

## Figures and Tables

**Figure 1 ijms-24-08090-f001:**
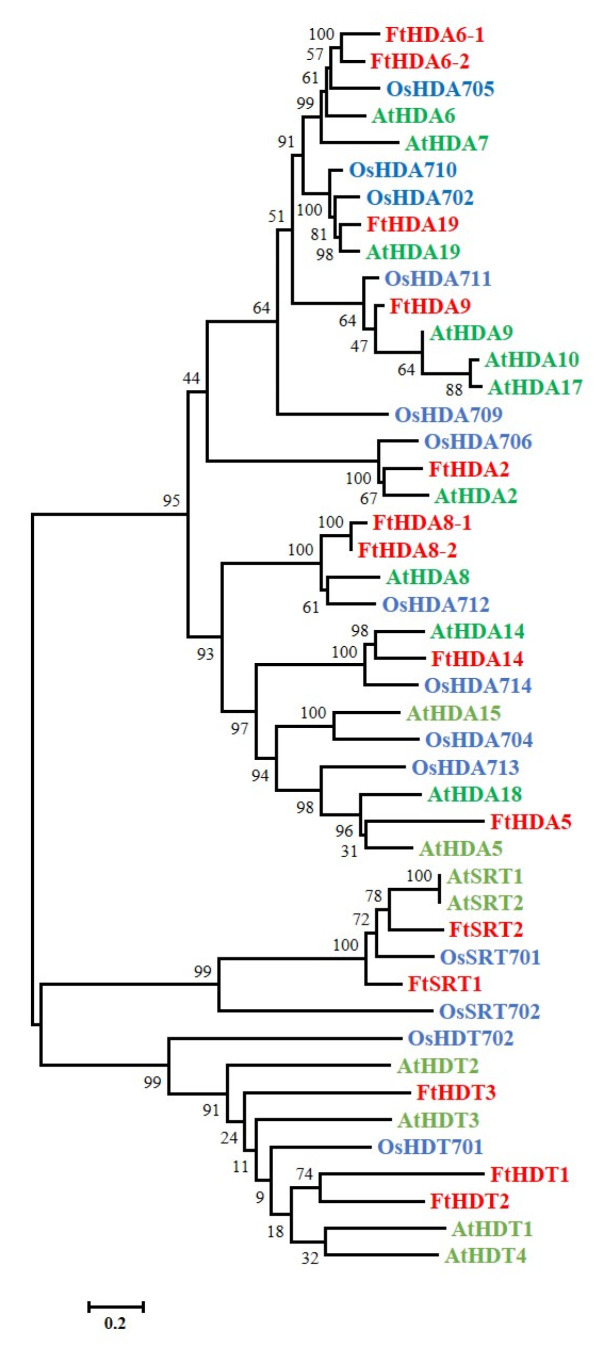
Phylogenetic analysis of HDAC proteins from *F. tataricum*, *Arabidopsis thaliana* and *Oryza sativa* (HDACs of *F. tataricum*: red; HDACs of Arabidopsis thaliana: green; HDACs of *Oryza sativa*: blue). The bar represents the number of substitutions per site.

**Figure 2 ijms-24-08090-f002:**
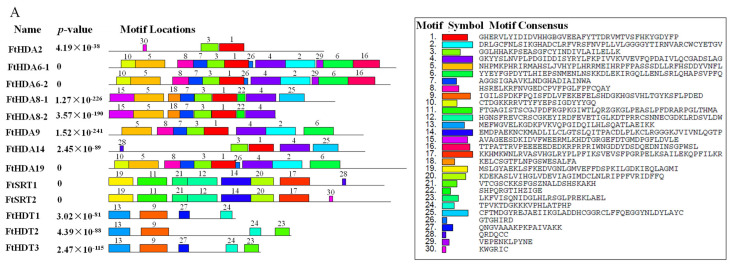
(**A**) Conserved motifs prediction of HDAC proteins in *F. tataricum* using the MEME program; (**B**) Conserved domains prediction of HDAC proteins in *F. tataricum* using SMART and HMMER softwares.

**Figure 3 ijms-24-08090-f003:**
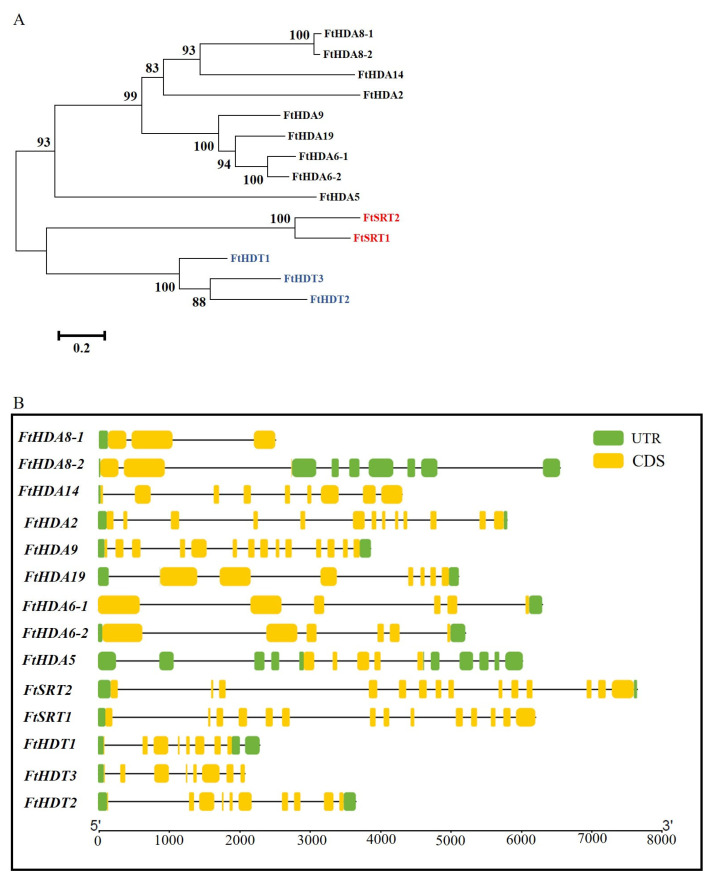
(**A**) The phylogenetic relationship of HDACs in *F. tataricum*. (RPD3/HDA1, black; SIR2, red; HD2, blue) The bar represents the number of substitutions per site. (**B**) Gene structure of HDAC genes in *F. tataricum*. Green boxes represent untranslated 5′- and 3′-regions; yellow boxes represent exons; black lines represent introns.

**Figure 4 ijms-24-08090-f004:**
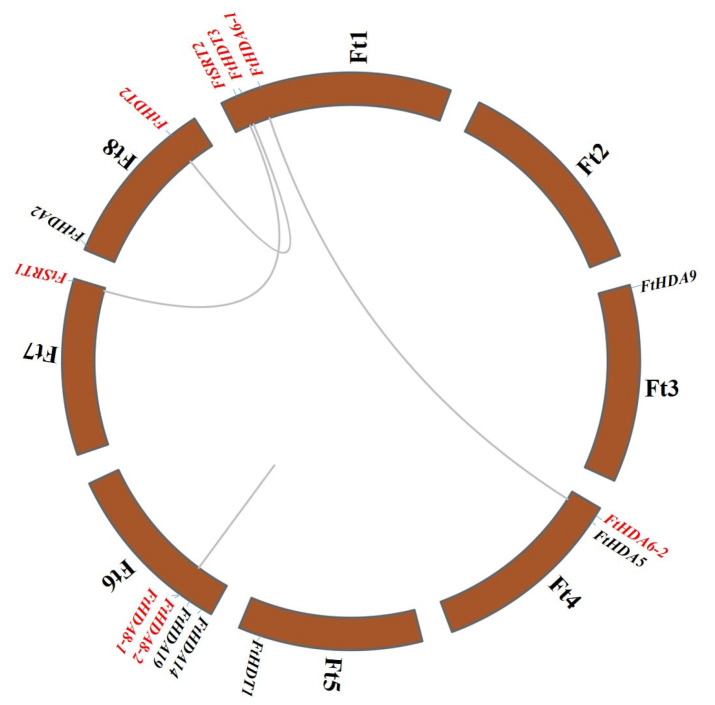
The distribution and duplication events of *FtHDACs* in *F. tataricum* genome. The gray lines connect the relevant 4 pairs of paralogous genes. Red: *FtHDACs* of paralogous gene pair.

**Figure 5 ijms-24-08090-f005:**
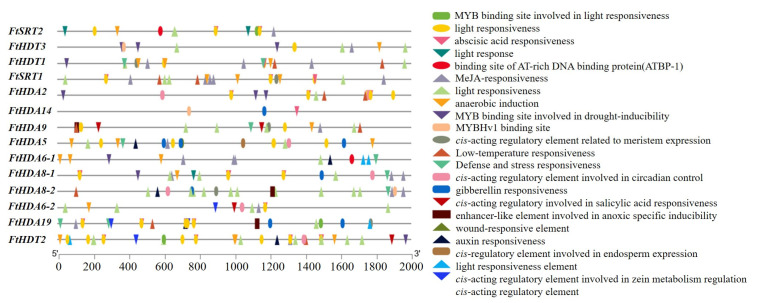
The *cis*-regulatory elements in the promoter regions of *FtHDACs*. The blocks in diverse color and shape indicate different elements.

**Figure 6 ijms-24-08090-f006:**
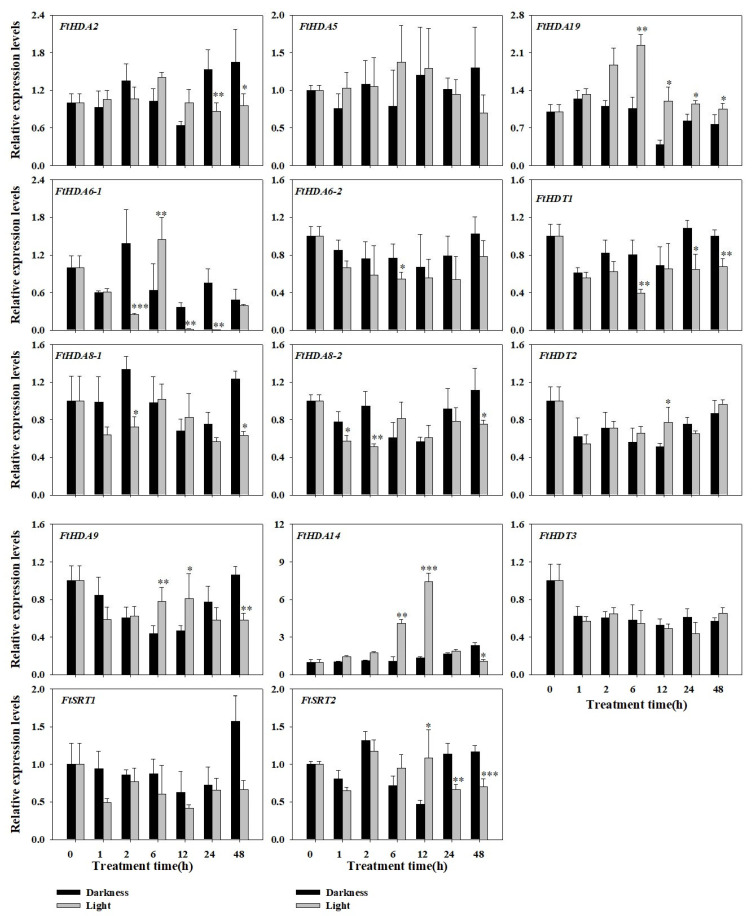
Expression profiles of 14 *FtHDAC* genes in response to light. *FtH3* was used as reference gene. Expression levels of each gene were expressed as a ratio relative to that of untreated seedlings (Darkness 0 h), which was set as 1. Each data point represents a mean ± standard error (*n* = 3). Asterisks above the bars indicate significant differences (* *p* < 0.05, ** *p* < 0.01, ****p* < 0.001).among the treatments.

**Figure 7 ijms-24-08090-f007:**
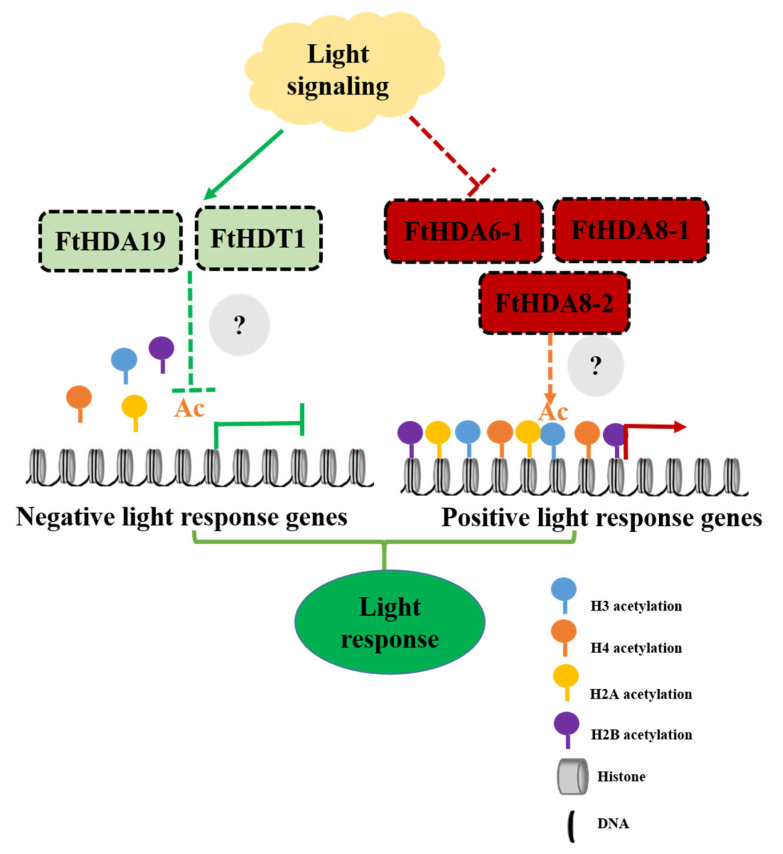
The proposed mechanism for the roles of FtHDACs in the light response of *F. tataricum*. *FtHDA19* and *FtHDT1* were induced by light treatment, which might reduce histone acetylation (Ac) of negative light responsive genes, thereby inhibiting their expression. *FtHDA6-1*, *FtHDA8-1,* and *FtHDA8-2* were inhibited by light treatment, which might result in higher levels of Ac in positive light responsive genes, thereby upregulating their expression. Other proteins might be involved in the recruitment of the histone deacetylases to light-responsive genes to regulate their expression.

**Table 1 ijms-24-08090-t001:** Overview of the histone deacetylases genes identified in *F. tataricum*.

Name	Gene ID	Chromosome Localization	CDS	Length (AA)	Mw	Name	Gene ID
*FtHDA2*	FtPinG0002233600.01	Ft8: 2509180–2515001	1092	363	39.98	8.44	AtHDA2
*FtHDA5*	FtPinG0004507000.01	Ft4: 4862314–4868357	552	183	20.68	9.39	AtHDA5
*FtHDA6-1*	FtPinG0005824600.01	Ft1: 11816747–11823065	1458	485	54.11	5.74	AtHDA6
*FtHDA6-2*	FtPinG0006680200.01	Ft4: 2793974–2799198	1428	475	53.65	5.39	AtHDA6
*FtHDA8-1*	FtPinG0006335400.01	Ft6: 10766017–10768527	1149	382	41.25	5.28	AtHDA8
*FtHDA8-2*	FtPinG0006335800.01	Ft6: 10775588–10782216	849	282	30.21	5.34	AtHDA8
*FtHDA9*	FtPinG0003240400.01	Ft3: 1572336–1576218	1293	430	49.19	5.06	AtHDA9
*FtHDA14*	FtPinG0002696900.01	Ft6: 3383062–3387385	1299	432	46.83	6.10	AtHDA14
*FtHDA19*	FtPinG0007059500.01	Ft6: 7283288–7288420	1509	502	56.19	5.08	AtHDA19
*FtSRT1*	FtPinG0001808100.01	Ft7: 50614511–50620732	1398	465	51.76	8.50	AtSRT1/2
*FtSRT2*	FtPinG0000318300.01	Ft1: 5066323–5073990	1431	476	52.75	9.35	AtSRT1/2
*FtHDT1*	FtPinG0001683500.01	Ft5: 47668125–47670424	645	214	23.55	4.11	AtHDT1/3/2
*FtHDT2*	FtPinG0007109100.01	Ft8: 41374550–41378216	927	308	33.35	4.67	AtHDT2/1/3
*FtHDT3*	FtPinG0001411500.01	Ft1: 6259288–6261377	771	256	27.83	5.10	AtHDT3/2/1

**Table 2 ijms-24-08090-t002:** The parameters and dates of the duplication events in the paralogous histone deacetylases genes of *F. tataricum*.

The Paralogous Gene		Ka	Ks	Ka/Ks	Date
					(Million Year Ago)
*FtHDA8-1*	*FtHDA8-2*	0.03	0.07	0.42	2.47
*FtHDA6-1*	*FtHDA6-2*	0.13	1.16	0.11	38.77
*FtHDT2*	*FtHDT3*	0.42	NaN	NaN	NaN
*FtSRT1*	*FtSRT2*	0.32	2.66	0.12	88.57

## Data Availability

Data are contained within the article.

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
