# Peer review of "Genome-Wide Identification of Histone Deacetylases and Their Roles Related with Light Response in Tartary Buckwheat (Fagopyrum tataricum)"

_ijms, 2023, doi:10.3390/ijms24098090_

Round 1
Reviewer 1 Report
Thank you for sending me this interesting manuscript by Huiling Yan et al. Overall, I judge this review as an interesting investigation describing 14 HDACs from F. tataricum. The authors study 14 HDACs, presenting information about Chromosomal localization, gene structure, conserved motifs, phylogenetic relationships, putative cis-elements in the promoters of FtHDAC genes. In addition, authors characterized expression patterns of FtHDACs in various tissues, fruits of different development stages and seedlings under different light conditions.
There are issues that are listed in order, as follow:
1) Page 3, Fig 1: Please include the model used to build the Phylogenic tree and what represents the bar.
2) page 4, Fig 2B: Please adjust the font size for the numbers underneath the sequence draw, they look too small.
3) page 8, line 225: “These results indicated that FtHDACs might involve in not only plant-environment interaction, but also phytohormone and stress signaling.” change to “These results indicated that FtHDACs might be involve in not only plant-environment interaction, but also phytohormone and stress signaling.”
4) page 12, line 363: “indicating it might involve in root and flower development” change to “indicating it might be involved in root and flower development”.
5) I would like to see housekeeping genes as control for the expression assays (actin, GAPDH) instead of H3.
6) add more information about histones/residues that are modified by HDACs.
7) for a better understanding of the HDACs target (histones), a figure would be included.
Author Response
Dear Editor and Reviewers,
Many thanks for the valuable comments and suggestions on our manuscript “Genome-wide identification of histone deacetylases and their roles related with light response in Tartary buckwheat (Fagopyrum tataricum)” (ID: ijms-2367275). Those comments are all valuable and very helpful for revising and improving our paper, as well as the important guiding significance for our research. We have revised the manuscript carefully and have made correction which we hope meet with approval. Any revisions made to the manuscript were marked up using the *Track Changes* function.
The following is the answers and revisions we have made in response to the reviewers’ questions and suggestions on an item by item basis.
Thank you for sending me this interesting manuscript by Huiling Yan et al. Overall, I judge this review as an interesting investigation describing 14 HDACs from F. tataricum. The authors study 14 HDACs, presenting information about Chromosomal localization, gene structure, conserved motifs, phylogenetic relationships, putative cis-elements in the promoters of FtHDAC genes. In addition, authors characterized expression patterns of FtHDACs in various tissues, fruits of different development stages and seedlings under different light conditions.
There are issues that are listed in order, as follow:
1) Page 3, Fig 1: Please include the model used to build the Phylogenic tree and what represents the bar.
Response 1: Thank you for your careful review and good suggestions. We have included the model information in method part and the information of the bar in the figure legends as you suggested.
2) page 4, Fig 2B: Please adjust the font size for the numbers underneath the sequence draw, they look too small.
Response 2: Thank you for your careful review and good suggestions. We have adjusted “Fig 2B” as you suggested.
3) page 8, line 225: “These results indicated that FtHDACs might involve in not only plant-environment interaction, but also phytohormone and stress signaling.” change to “These results indicated that FtHDACs might be involve in not only plant-environment interaction, but also phytohormone and stress signaling.”
Response 3: Thank you for your careful review and good suggestions. We have revised the sentence as you suggested.
4) page 12, line 363: “indicating it might involve in root and flower development” change to “indicating it might be involved in root and flower development”.
Response 4: Thank you for your careful review and good suggestions. We have revised the sentence as you suggested.
5) I would like to see housekeeping genes as control for the expression assays (actin, GAPDH) instead of H3.
Response 5: Thank you for your constructive suggestion. We have added FtACTIN7 as the other internal reference gene and we put the result of RT-qPCR detection as Fig.S1 in Appendix.
6) add more information about histones/residues that are modified by HDACs.
Response 6: Thank you for your constructive suggestion. We have supplemented the relative information in the introduction part as you suggested.
7) for a better understanding of the HDACs target (histones), a figure would be included.
Response 7: Thank you for your constructive suggestion. We have supplemented the HDACs target information in Figure 9 and hope it meets your approval.
Thank you very much for your kindly consideration and reviewer’s comments and suggestions.
Best Regards,
Yours sincerely,
Jianglin Zhao
Key Laboratory of Coarse Cereal Processing, Ministry of Agriculture and Rural Affairs, Sichuan Province Engineering Technology Research Center of Coarse Cereal Industralization, School of Food and Biological Engineering, Chengdu University, Chengdu 610106, China
E-mail: jlzhao@cdu.edu.cn (J. Zhao)
Reviewer 2 Report
Dear Authors,
This is a very nice manuscript.
Please find my comments in the attached file.
Best regards

Dear Authors,
The quality of the English Language is good.
However I detected some issues throughout the manuscript.
Please find my comments in the attached file.
Best regards
Author Response
Dear Editor and Reviewers,
Many thanks for the valuable comments and suggestions on our manuscript “Genome-wide identification of histone deacetylases and their roles related with light response in Tartary buckwheat (Fagopyrum tataricum)” (ID: ijms-2367275). Those comments are all valuable and very helpful for revising and improving our paper, as well as the important guiding significance for our research. We have revised the manuscript carefully and have made correction which we hope meet with approval. Any revisions made to the manuscript were marked up using the *Track Changes* function.
The following is the answers and revisions we have made in response to the reviewers’ questions and suggestions on an item by item basis.
- Dear Authors,
This is a very nice manuscript.
Please find my comments in the attached file.
Best regards
Response 1: Thank you for your careful review and good suggestions. We have revised the manuscript as you suggested.
- Dear Authors,
The quality of the English Language is good.
However I detected some issues throughout the manuscript.
Please find my comments in the attached file.
Best regards
Response 2: Thank you for your careful review and good suggestions. We have revised all the issues as you suggested.
- please write the latin name altogether (in the third line) and not divided like that
Response 3: Thank you for your careful review and good suggestions. We have revised the latin name as you suggested.
- please follow the format of the paper for all names in this section
Response 4: Thank you for your careful review. We have revised the part as you suggested.
5) please provide the latin name, the authorities, order and family
Response 5: Thank you for your careful review and good suggestions. We have revised the part as you suggested.
6) what are these three groups, please explain
Response 6: Thank you for your careful review and good suggestions. We have explained the groups as you suggested.
7) please add "The" in the beginning of the sentence
Response 7: Thank you for your careful review and good suggestions. We have added “The” as you suggested.
8) please add references
Response 8: Thank you for your careful review and good suggestions. We have added the reference as you suggested.
9) please delete the "spaces" between the numbers, i.e., [1,3,4]. Please change throughout the manuscript
Response 9: Thank you for your careful review. We have revised the format throughout the manuscript as you suggested.
10) please use the "–" and not "-". Please replace throughout the manuscript and the references
Response 10: Thank you for your careful review. We have revised the format throughout the manuscript as you suggest.
11) please put a comma after Tartary buckwheat, remove the parenthesis and add authorities, after the latin name
Response 11: Thank you for your careful review. We have revised the part the manuscript as you suggest.
12) please delete and write "F. tataricum".Please replace throughout the manuscript
Response 12: Thank you for your careful review. We have revised it the part throughout the manuscript as you suggest.
13) please add references
Response 13: Thank you for your constructive suggestions. We have added references as you suggest.
14) please add "The" in the beginning of the sentence
Response 14: Thank you for your careful review and good suggestions. We have added “The” as you suggested.
15) please provide latin name, authorities, order and family
Response 15: Thank you for your careful review and good suggestions. We have revised it as you suggest.
16) please write the latin name
Response 16: Thank you for your careful review and good suggestions. We have revised it as you suggest.
17) please add references
Response 17: Thank you for your constructive suggestions. We have added references as you suggest.
18) please italics
Response 18: Thank you for your careful review. We have revised it as you suggest.
19) please write here the objectives of your study and not the results.
Response 19: Thank you for your careful review and good suggestions. We have revised it as you suggest.
20) please provide latin names, authorities, orders and families of each species.
Response 20: Thank you for your careful review and good suggestions. We have revised them as you suggest.
21) please add these in the reference section and not like this into the text
Response 21: Thank you for your careful review and good suggestions. We have revised them as you suggest.
22) please add references.
Response 22: Thank you for your careful review and good suggestions. We have added references as you suggest.
23) please provide authorities, order and family of each species
Response 23: Thank you for your careful review and good suggestions. We have revised them as you suggest.
24) please use the latin names not the common names
Response 24: Thank you for your careful review and good suggestions. We have replaced them with latin names as you suggest.
25) please use latin names
Response 25: Thank you for your careful review and good suggestions. We have replaced them with latin names as you suggest.
26) Please add "The" in the beginning of the sentence
Response 26: Thank you for your careful review and good suggestions. We have added “The” as you suggested.
27) Please add "The" in the beginning of the sentence
Response 27: Thank you for your careful review and good suggestions. We have added “The” as you suggested.
28) Please add "The" in the beginning of the sentence
Response 28: Thank you for your careful review and good suggestions. We have added “The” as you suggested.
29) Please add "The" in the beginning of the sentence
Response 29: Thank you for your careful review and good suggestions. We have added “The” as you suggested.
30) Please add "The" in the beginning of the sentence
Response 30: Thank you for your careful review and good suggestions. We have added “The” as you suggested.
31) Please add "The" in the beginning of the sentence
Response 31: Thank you for your careful review and good suggestions. We have added “The” as you suggested.
32) Please add "The" in the beginning of the sentence
Response 32: Thank you for your careful review and good suggestions. We have added “The” as you suggested.
33) please delete this part, you write that again in the beginning of the manuscript
Response 33: Thank you for your careful review and good suggestions. We have revised it as you suggested.
34) please delete this and the parenthesis
Response 34: Thank you for your careful review and good suggestions. We have revised it as you suggested.
35) Please add "The" in the beginning of the sentence
Response 35: Thank you for your careful review and good suggestions. We have added “The” as you suggested.
36) Please add "The" in the beginning of the sentence
Response 36: Thank you for your careful review and good suggestions. We have added “The” as you suggested.
37) Please add "The" in the beginning of the sentence
Response 37: Thank you for your careful review and good suggestions. We have added “The” as you suggested.
38) Please add "The" in the beginning of the sentence.
Response 38: Thank you for your careful review and good suggestions. We have added “The” as you suggested.
39) Please add references.
Response 39: Thank you for your careful review and good suggestions. We have added references as you suggest.
40) Please use the latin names.
Response 40: Thank you for your careful review and good suggestions. We have revised them to latin names as you suggest.
41) Please write these as references and not like that in the text.
Response 41: Thank you for your careful review and good suggestions. We have revised them as you suggest.
42) Please write these as references and not like that in the text.
Response 42: Thank you for your careful review and good suggestions. We have revised them as you suggest.
43) please use latin names.
Response 43: Thank you for your careful review and good suggestions. We have revised them with latin names as you suggest.
44) please write these as references and not like that in the text.
Response 44: Thank you for your careful review and good suggestions. We have revised them as you suggest.
45) please write these as references and not like that in the text.
Response 45: Thank you for your careful review and good suggestions. We have revised them as you suggest.
46) please write these as references and not like that in the text.
Response 46: Thank you for your careful review and good suggestions. We have revised them as you suggest.
47) please add "The" in the beginning of the sentence.
Response 47: Thank you for your careful review. We have added “The” as you suggest.
48) please follow the format of the journal throughout the references.
Response 48: Thank you for your careful review. We have revised all the references according to the format of the journal as you suggest.
Thank you very much for your kindly consideration and reviewer’s comments and suggestions.
Best Regards,
Yours sincerely,
Jianglin Zhao
Key Laboratory of Coarse Cereal Processing, Ministry of Agriculture and Rural Affairs, Sichuan Province Engineering Technology Research Center of Coarse Cereal Industralization, School of Food and Biological Engineering, Chengdu University, Chengdu 610106, China
E-mail: jlzhao@cdu.edu.cn (J. Zhao)